# Trip-ROMA: Self-Supervised Learning with Triplets and Random Mappings

Wenbin Li[1], Xuesong Yang[1], Meihao Kong[1], Lei Wang[2], Jing Huo[1], Yang Gao[1]*, Jiebo Luo[3]
[1]**State Key Laboratory for Novel Software Technology, Nanjing University, China**
[2]**University of Wollongong, Australia**, [3]**University of Rochester, USA**

**Reviewed on OpenReview:** `https://openreview.net/forum?id=MR4glug5GU`

## Abstract

Contrastive self-supervised learning (SSL) methods, such as MoCo and SimCLR, have achieved great success in unsupervised visual representation learning. They rely on a large number of negative pairs and thus require either large memory banks or large batches. Some recent non-contrastive SSL methods, such as BYOL and SimSiam, attempt to discard negative pairs and have also shown remarkable performance. To avoid collapsed solutions caused by not using negative pairs, these methods require non-trivial asymmetry designs. However, in small data regimes, we can not obtain a sufficient number of negative pairs or effectively avoid the over-fitting problem when negatives are not used at all. To address this situation, we argue that negative pairs are still important but one is generally sufficient for each positive pair. We show that a simple *Triplet-based loss (Trip)* can achieve surprisingly good performance without requiring large batches or asymmetry designs. Moreover, to alleviate the over-fitting problem in small data regimes and further enhance the effect of *Trip*, we propose a simple plug-and-play *RandOm MApping (ROMA)* strategy by randomly mapping samples into other spaces and requiring these randomly projected samples to satisfy the same relationship indicated by the triplets. Integrating the triplet-based loss with random mapping, we obtain the proposed method *Trip-ROMA*. Extensive experiments, including unsupervised representation learning and unsupervised few-shot learning, have been conducted on ImageNet-1K and seven small datasets. They successfully demonstrate the effectiveness of Trip-ROMA and consistently show that ROMA can further effectively boost other SSL methods. Code is available at `https://github.com/WenbinLee/Trip-ROMA`.

## 1 Introduction

Unsupervised visual representation learning aims to learn good image representations without human supervision. To achieve this, self-supervised learning (SSL) has received considerable attention and shown promise in recent years (Wu et al., 2018; He et al., 2020; Chen et al., 2020a; Chen & He, 2021; Grill et al., 2020; Tian et al., 2020; Caron et al., 2020; Zbontar et al., 2021). The recent advances can be classified into two categories (Tian et al., 2021): contrastive SSL and non-contrastive SSL. Contrastive SSL, such as Sim-CLR (Chen et al., 2020a) and MoCo (He et al., 2020; Chen et al., 2020b), learns representations by closing the latent representations of two views of the same image together (positive pairs), while pushing the latent representations of different images farther away (negative pairs or inter-image). These methods normally rely on a large number of negative pairs by using large batches or memory banks. Recently, non-contrastive SSL, such as BYOL (Grill et al., 2020) and SimSiam (Chen & He, 2021), has attempted to learn representations without using negative pairs. To avoid collapsed representations caused by removing negative pairs, these methods usually have to employ some non-trivial asymmetry designs, such as asymmetric predictor

---
*Corresponding author.

network (Grill et al., 2020; Chen & He, 2021), stop-gradients (Grill et al., 2020; Chen & He, 2021), and momentum encoder (Grill et al., 2020). However, in the small data regimes, *e.g.,* unsupervised few-shot learning, it is not able to use a large number of negative pairs, and it is also difficult to avoid the over-fitting problem in the case of small data setting, especially when the negative pairs are discarded at all. Therefore, this raises an interesting question – "*How to effectively utilize negatives in self-supervised learning, especially on small data?*"

In addition, to learn view-invariant representations, both contrastive and non-contrastive SSL normally use data augmentation to obtain pseudo labels, considering that data augmentation can well maintains the semantics of the original examples. Unfortunately, such an advantage could make models be prone to the over-fitting problem during training. The similar point has also been mentioned in SimCLR (Chen et al., 2020a) and InfoMin (Tian et al., 2020), which show that stronger data augmentation or reducing the mutual information between augmented views of the same image can bring more benefits. In fact, the asymmetric designs in non-constrastive SSL can also be regarded as a way to alleviate the over-fitting problem. Unfortunately, this over-fitting problem will be even worse on small data where the number of training samples is scarce. This naturally raises another question – "*Do we have more efficient ways to alleviate the over-fitting problem in data augmentation based SSL methods?*"

In this paper, we propose a new method, *Trip-ROMA*, as responses to the above two questions. For the first question, we find that *even when the number of negative pairs is reduced to one, i.e., just one negative, it will be generally sufficient for each positive pair (i.e., triplet)*. In the literature, some methods (*e.g.,* SimCLR (Chen et al., 2020a)) have ever tried to use the margin triplet loss (Schroff et al., 2015) based on triplets, but did not achieve satisfied results. In this work, we revisit the effect of triplets, but differently we employ a *triplet + binary cross-entropy loss* built on triplets constructed within each minibatch for training. Compared with other methods, such a simple *triplet-based loss (Trip)* requires neither large batches (Chen et al., 2020a) nor memory banks (He et al., 2020), which is therefore more suitable for small data. Also, it does not require any asymmetric design like asymmetric predictor network (Grill et al., 2020; Chen & He, 2021) or stop-gradients (Grill et al., 2020; Chen & He, 2021). In contrast, it enjoys advantages of both existing contrastive and non-contrastive SSL methods: (1) working well with small batches, (2) learning from both intra- and inter-image information, and (3) naturally avoiding collapsed representations.

As for the second question, we propose a "*random mapping preemptive measure*" into SSL to help unsupervised deep models efficiently deal with over-fitting. In other words, we can learn more robust representations by forcing models maintain the required sample relationship even under random mappings. This is because sometimes the relationship of samples during the training process could be satisfied by chance or over-fitting. Taking *RandOm MApping (ROMA)* as a preemptive measure can alleviate such a problem to some extent, and thus bring a more robust way to evaluate the relationship between samples. In addition, random mappings could enhance the diversity of samples' features, which can also alleviate the over-fitting caused by the scarce of samples in the small data regimes. In this work, we provide interpretation to the effect of the proposed random mapping strategy from the perspective of perturbation of the coordinate system. Because the same relationship of samples in triplets is required to be satisfied for all random mappings, we name our method *Trip-ROMA*, taking the meaning from *"all roads lead to a trip to Rome"*.

Typically, the existing SSL methods are mainly designed on large-scale data, *e.g.,* ImageNet-1K (Deng et al., 2009), while paying less attention to small data. Therefore, in this work, we mainly focus on small data and investigate the transferability of SSL models on small data. Through extensive experiments, we show that our Trip-ROMA can achieve a new state of the art on multiple benchmarks. Specifically, under the linear evaluation protocol, Trip-ROMA achieves 83.05% top-1 accuracy on ImageNet-100, which is a 3.07% absolute improvement over SimCLR (Chen et al., 2020a). Using the $k$-NN evaluation protocol, Trip-ROMA achieves 58.02% top-1 accuracy on *mini*ImageNet under the 5-way 1-shot few-shot setting, which is a 5.26% absolute improvement over SimSiam (Chen & He, 2021). Moreover, when extended to the large-scale dataset, Trip-ROMA can also obtain a superior top-1 accuracy of 71.1% on ImageNet-1K under the linear evaluation protocol compared to the state-of-the-art methods. In addition, as a plug-and-play strategy, the ROMA strategy can also consistently improve the performance of existing SSL methods.

Our main contributions of this work are summarized as follows:

- We demonstrate that negative pairs are still important but one negative pair is generally sufficient for each positive pair in contrastive SSL, especially on small data. A simple and effective triplet-based loss (*Trip*) is designed accordingly to validate this approach.

- We propose a novel plug-and-play random mapping strategy, *ROMA*, to enforce SSL models to learn under random mappings and demonstrate that unsupervised representation learning can effectively benefit from incorporating randomness.

- We experimentally demonstrate that the proposed *Trip-ROMA* achieves new state of the art on both unsupervised representation learning and few-shot learning.

## 2 Related Work

**Contrastive learning.** Recently, contrastive learning based SSL (Bachman et al., 2019; Ye et al., 2019; Hjelm et al., 2019; Misra & Maaten, 2020; Oord et al., 2018; Wu et al., 2018; He et al., 2020; Chen et al., 2020a; Hénaff et al., 2020; Chen et al., 2020b; Tian et al., 2019; 2020; Wei et al., 2021; Xie et al., 2021) has drawn increasing attention owing to its simple design and excellent performance. The core idea is to discriminate among different images, by maximizing the similarity between two augmented views of the same image and repulsing other different images, *i.e.,* contrastive learning (Hadsell et al., 2006). Along this way, Dosovitskiy et al. (2014) and Wu et al. (2018) propose to take each patch/image as an individual class via instance-level discrimination. MoCo (He et al., 2020) and MoCo v2 (Chen et al., 2020b) match an encoded query to a dictionary of encoded keys with a slow-moving average network using an InfoNCE loss (Hénaff et al., 2020). ReSSL (Zheng et al., 2021) proposes a relational SSL framework to model the relationship between different images, which is built on MoCo v2. Differently, SimCLR (Chen et al., 2020a) directly constructs negative samples within a much larger minibatch without requiring additional memory banks. Instead of using an instance-level contrastive task, Pixpro (Xie et al., 2021) proposes a pixel-to-propagation consistency pretext task from the perspective of pixel level. Although this type of methods performs excellently, they usually requires a large number of negative examples to work well, by using either memory banks or large batches.

**Non-contrastive learning.** Different from the above contrastive learning based SSL, recent literature has attempted to only use positive pairs and completely discard the negative examples, named non-contrastive learning based SSL (Caron et al., 2020; Grill et al., 2020; Chen & He, 2021; Zbontar et al., 2021). BYOL (Grill et al., 2020) directly predicts the representation of one augmented view from another view of the same image with a Siamese network (Bromley et al., 1993), where one branch of the network is a slow-moving average (momentum encoder) of another branch. After that, based on BYOL, RSA (Bai et al., 2022) proposes a new multi-stage augmentation pipeline to reduce the severe semantic shift of using aggressive augmentations. SimSiam (Chen & He, 2021) discards the momentum encoder, directly maximizes the similarity between two augmented views of one image, and demonstrates that the stop-gradient operation is critical for preventing collapsing. W-MSE (Ermolov et al., 2021) scatters all the sample representations into a spherical distribution using a whitening transform and penalizes the positive pairs that are far from each other to avoid the collapse problem. In addition, SwAV (Caron et al., 2020) performs online clustering to learn prototypes and indirectly compares two augmented views of the same image by using their cluster assignments built on the learned prototypes. Recently, Barlow Twins (Zbontar et al., 2021) maximizes the similarity between two augmented views of one image while reducing redundancy between their components, by relying on very high-dimensional representations. Although these methods successfully discard the negative examples and thus somewhat alleviate the computation, they introduce a new collapse problem and require further effort to address this new problem.

As mentioned above, the existing contrastive SSL is confined to requiring massive negative examples, while the non-contrastive SSL easily suffers from the collapse problem. As a compromise, Trip-ROMA employs one negative example to naturally avoid the collapse problem. Importantly, although only one negative example is used, Trip-ROMA can still achieve remarkable performance. In addition, a new plug-and-play random mapping strategy is proposed, which can further boost the unsupervised representation learning.

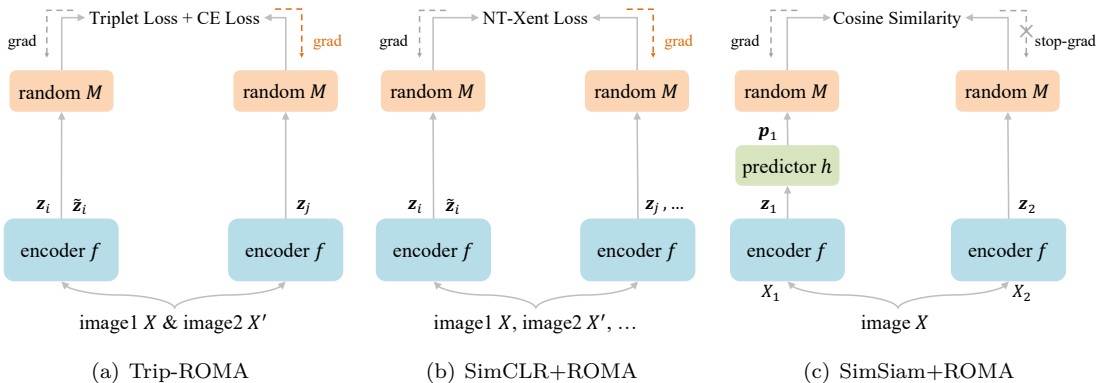

Figure 1: (a) Trip-ROMA architecture with a triplet+cross-entropy (CE) loss. (b) SimCLR architecture introducing the random mapping strategy, named SimCLR+ROMA. (c) SimSiam architecture using the random mapping strategy, named SimSiam+ROMA. All three models employ the same encoder $f$ (a backbone plus a projection MLP) in both branches except SimSiam uses an additional predictor $h$ in one branch. For each input positive image, SimSiam needs no negative examples, SimCLR requires a large number of negative examples, while Trip-ROMA only needs one negative example. A random PSD matrix $M \succeq 0$ is used to calculate the final loss, which will be discarded during test.

## 3 The Proposed Method

### 3.1 Trip: Triplet-based Loss Function

Following the existing contrastive and non-contrastive SSL methods (He et al., 2020; Chen et al., 2020a; Chen & He, 2021; Grill et al., 2020; Tian et al., 2020; Caron et al., 2020; Zbontar et al., 2021), we use data augmentation to obtain the image-level pseudo labels, *i.e.,* each image is regarded as one individual class. As shown in Fig. 1(a), given any two different images $\langle X, X' \rangle$ within a minibatch, we can construct a triplet $\langle X_i, \tilde{X}_i, X_j \rangle$ via data augmentation, where both $X_i$ and $\tilde{X}_i$ (a positive pair) are augmented from $X$, and $X_j$ is augmented from $X'$ (a negative example). After that, this triplet will be processed by an encoder network $f$ that consists of a backbone and a projection multilayer perceptron (MLP), obtaining the corresponding feature vectors $\langle z_i, \tilde{z}_i, z_j \rangle, z \in \mathbb{R}^d$. A simple *triplet + binary cross-entropy loss* for a triplet can be formulated as

$$\mathcal{L}_{ij} = \left[ z_i^\top z_j - z_i^\top \tilde{z}_i + 1 \right]_+ - \lambda \cdot \log \frac{\exp(z_i^\top \tilde{z}_i / \tau)}{\exp(z_i^\top \tilde{z}_i / \tau) + \exp(z_i^\top z_j / \tau)} , \tag{1}$$

where $z_i$, $\tilde{z}_i$ and $z_j$ have been $\ell_2$-normalized, $[z]_+ = \max(0, z)$, $\lambda$ is a hyper-parameter trading off the importance of the two loss terms, and $\tau$ is a temperature parameter.

As seen, the first term of Eq. (1) is a triplet loss (Weinberger & Saul, 2009), and the second term is a cross-entropy loss for binary classification. In general, the triplet loss will suffer from a limitation of being sensitive to the training triples because of using a fixed margin (Manmatha et al., 2017). Therefore, the cross-entropy loss can be seen as a *soft triplet loss* but with an adaptive margin, compensating for the limitation of the triplet loss with a fixed margin. Note that each part alone of Eq. (1) is not new and has been used in previous work (Parkhi et al., 2015; Koch et al., 2015; Wu et al., 2018; He et al., 2020; Chen et al., 2020a; Tian et al., 2020). Nevertheless, our key contribution here is to demonstrate that such a combination of both losses can achieve surprisingly good performance in SSL on small data. This has not been clearly shown in the literature and it reveals that negative pairs are still important but one may be sufficient.

### 3.2 ROMA: Similarity Measure with Randomness

The existing contrastive and non-contrastive SSL methods normally use cosine similarity $Sim(\boldsymbol{u}, \boldsymbol{v}) = \boldsymbol{u}^\top \boldsymbol{v} / \|\boldsymbol{u}\|_2 \|\boldsymbol{v}\|_2$ to calculate the similarity between points $\boldsymbol{u} \in \mathbb{R}^d$ and $\boldsymbol{v} \in \mathbb{R}^d$. During the optimization

process of training, the relationship of points through such a measure could be satisfied by chance or due to overfitting. An interesting question is whether we can slightly modify this measure to deal with this situation so as to obtain a more robust evaluation on the relationship of points. Our answer is yes. To be specific, we assume both $\boldsymbol{u}$ and $\boldsymbol{v}$ have been $\ell_2$-normalized for simplicity. Then $Sim(\boldsymbol{u}, \boldsymbol{v})$ reduces to $\boldsymbol{u}^\top \boldsymbol{v}$ and it can be expressed as $Sim(\boldsymbol{u}, \boldsymbol{v}) = \boldsymbol{u}^\top I \boldsymbol{v}$, where $I$ is the identity matrix. To obtain a more robust evaluation on the relationship between $\boldsymbol{u}$ and $\boldsymbol{v}$, we replace $I$ with a random positive semi-definite (PSD) matrix $M \in \mathbb{R}^{d \times d}, M \succeq 0$ as follows,

$$Sim(\boldsymbol{u}, \boldsymbol{v}) = \boldsymbol{u}^\top M \boldsymbol{v}. \tag{2}$$

In this way, we can conveniently introduce randomness into this similarity measure and then further into the objective function in Eq. (1), by using a series of random matrices $\{M_k\}|_{k=1}^n, M_k \in \mathbb{R}^{d \times d}$ ($n$ is the number of random matrices). More specifically, when using the proposed similarity measure in Eq. (2), the loss in Eq. (1) can be rewritten as

$$\mathcal{L}_{ij}^{\text{random}} = \left[ \boldsymbol{z}_i^\top M \boldsymbol{z}_j - \boldsymbol{z}_i^\top M \tilde{\boldsymbol{z}}_i + 1 \right]_+ - \lambda \cdot \log \frac{\exp(\boldsymbol{z}_i^\top M \tilde{\boldsymbol{z}}_i / \tau)}{\exp(\boldsymbol{z}_i^\top M \tilde{\boldsymbol{z}}_i / \tau) + \exp(\boldsymbol{z}_i^\top M_i \boldsymbol{z}_j / \tau)}. \tag{3}$$

As will be demonstrated in the experimental study, the adoption of the random matrix $M$ consistently leads to better feature representation after the learning process. To gain better understanding on the use of $M$, we interpret its effect from the following perspective of perturbation.

**Perturbation of the coordinates.** We explain the effect of using the random matrix $M$ from a perspective of linear projection, *i.e.,* perturbing the coordinate system $\mathbb{R}^d$ with random rotation and scaling. In particular, because $M$ is positive semi-definite (PSD) in Eq. (2), $M$ can be mathematically decomposed as $L^\top L$, where $L \in \mathbb{R}^{c \times d}$ where $c$ is the rank of $M$ ($c \leq d$). After that, Eq. (2) can be converted as

$$Sim(\boldsymbol{u}, \boldsymbol{v}) = \boldsymbol{u}^\top M \boldsymbol{v} = \boldsymbol{u}^\top L^\top L \boldsymbol{v} = (L\boldsymbol{u})^\top (L\boldsymbol{v}), \tag{4}$$

where $L$ is a random linear transformation matrix, randomly mapping the data points into a new (perturbed) coordinate system. In the new system, cosine similarity is then evaluated. By employing multiple different transformation matrices in Eq. (4), we hope to obtain a more reliable evaluation on the relationship of the two points. In this sense, we essentially advocate a "*random mapping preemptive measure*" for SSL. That is, to avoid the relationship among points from being satisfied by chance or due to overfitting, we shall take a more aggressive approach to evaluate their relationship, *i.e.,* a stable similarity or dissimilarity relationship shall more likely survive random perturbation.

Considering that directly generating a random PSD matrix $M$ is difficult due to the requirement of the property of positive semi-definiteness, we instead randomly generate the linear projection matrix $\{L_k\}|_{k=1}^n$, making each $L_k^\top L_k$ naturally positive semi-definite. The illustration of random mappings for a triplet is shown in Fig. 2. Note that for each triplet in a minibatch, we only use a single random linear projection matrix $L_k$. Also, the frequency of using the random mapping will be discussed in Section 5.3 of experiments.

### 3.3 Generality of ROMA

We highlight that *RandOm MApping (ROMA)* is a general strategy. In other words, as a plug-and-play strategy, ROMA can be applied to other state-of-the-art SSL methods, no matter they are contrastive based or non-contrastive based. Taking SimCLR (Chen et al., 2020a) as an example, when the random mappings are incorporated, named *SimCLR+ROMA*, its random-based NT-Xent loss can be easily formulated as

$$\mathcal{L}^{\text{SimCLR+ROMA}} = -\log \frac{\exp(\boldsymbol{z}_i^\top M \tilde{\boldsymbol{z}}_i / \tau)}{\sum_{j=1}^{2N} \mathbb{I}_{j \neq i} \exp(\boldsymbol{z}_i^\top M \boldsymbol{z}_j / \tau)}, \tag{5}$$

where $\mathbb{I}_{k \neq i}$ is an indicator function, $N$ is the number of batch size, and $M$ is a random PSD matrix. Its architecture can be seen in Fig. 1(b).

Similarly, introducing the random mappings into SimSiam (Chen & He, 2021), we can obtain *SimSiam+ROMA* and its symmetrized cosine similarity loss can be easily converted as

$$\mathcal{L}^{\text{SimSiam+ROMA}} = -\frac{1}{2} \boldsymbol{p}_1^\top M \boldsymbol{z}_2 - \frac{1}{2} \boldsymbol{p}_2^\top M \boldsymbol{z}_1, \tag{6}$$

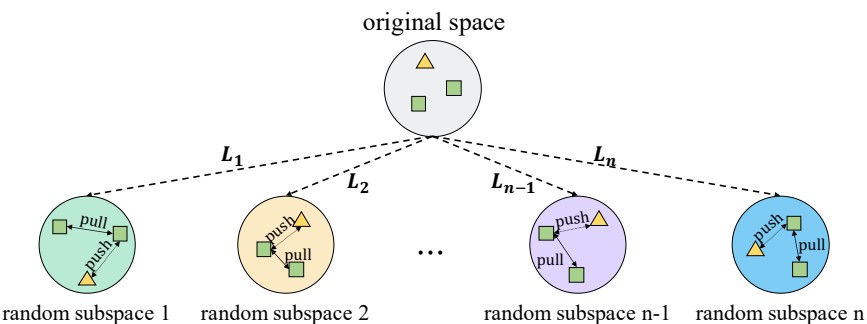

Figure 2: Illustration of random mappings for a triplet. For each triplet, the original feature space can be projected into another feature space with a random matrix $L_i$. In each random subspace, the relevant structure of the triplet, *i.e.,* the similarity between the positive pair should be larger than the negative pair, will be enforced by pulling and pushing operations. Rectangles and triangles denote positive and negative samples, respectively. Note that for one certain minibatch, only one single random matrix $L_i$ is used.

where $\langle z_1, z_2 \rangle$ is a positive pair, outputs of the encoder $f$, and $\langle p_1, p_2 \rangle$ is the subsequent representations of $\langle z_1, z_2 \rangle$ by further processing via an additional predictor $h$. See Fig. 1(c) for a more intuitive understanding. In addition, both SimCLR+ROMA and SimSiam+ROMA will be further investigated and discussed in the experimental part. Please see Section 5.1 for more details.

## 4    Implementation Details

**Datasets.** The main experiments are conducted on seven small benchmark datasets, *i.e.,* CIFAR-10 (Krizhevsky et al., 2009), CIFAR-100 (Krizhevsky et al., 2009), STL-10 (Coates et al., 2011), ImageNet-100 (Tian et al., 2019), *mini*ImageNet (Vinyals et al., 2016), CIFAR-100FS (Bharti et al., 2020) and FC100 (Oreshkin et al., 2018), whose details can be found in the appendix.

**Image augmentation.** For a fair comparison, we use the same data augmentation operations as Sim-CLR (Chen et al., 2020a), consisting of random cropping and resizing, horizontal flipping, color distortion, Gaussian blurring, and an optional grayscale conversion. Please refer to SimCLR (Chen et al., 2020a) for more details.

**Architecture.** We utilize *ResNet-50* (He et al., 2016), *ResNet-18* and *ResNet-12* (Lee et al., 2019) followed by a MLP-based projector network as our encoder $f$, respectively, where *ResNet-12* is used for the unsupervised few-shot learning (FSL) task. For the generic unsupervised representation learning task, the MLP-based projector network consists of three fully-connected (FC) layers, each with 2048 output dimensions. Specifically, the first two FC layers are followed by a batch normalization (BN) layer (Ioffe & Szegedy, 2015) and a leaky rectified linear unit (LeakyReLU) layer (Maas et al., 2013), while only the BN layer is used in the last FC layer. As for the unsupervised FSL task, there are only two FC layers in the MLP-based projector. Furthermore, different from the mainstream methods (Chen et al., 2020a; Grill et al., 2020; Chen & He, 2021), we add a random projection matrix with size of $2048 \times 1024$, *i.e.,* $L \in \mathbb{R}^{2048 \times 1024}$, after the encoder module to learn more robust representations.

**Optimization.** We use SGD optimizer with a momentum of 0.9 and a cosine decay learning rate schedule to optimize Trip-ROMA. On ImageNet-100, we train Trip-ROMA for 200 epochs with the base learning rate of 0.2, batch size of 128 and weight decay of $1e-4$. For CIFAR-10, STL-10, CIFAR-100 and *mini*ImageNet, the models are trained for 1000 epochs with the base learning rate of 0.03 and weight decay of $5e-4$. The batch size is set to 64 for CIFAR-10 and CIFAR-100, but 128 for STL-10, *mini*ImageNet, CIFAR-100FS and FC100. Additionally, each learning rate is linearly scaled with the batch size. The hyper-parameters $\lambda$ and $\tau$ in Eq. (3) are set to 8 and 0.5, respectively. For the random mapping setting, a linear transformation matrix $L$ with size of $2048 \times 1024$ is randomly sampled from a standard normal distribution.

**Linear evaluation.** Following the SSL literature (Zhang et al., 2016; Oord et al., 2018; Chen et al., 2020a; Grill et al., 2020; Chen & He, 2021), we employ a linear evaluation protocol to evaluate the features learned by our model for the unsupervised representation learning task. To be specific, given the pre-trained ResNet backbone after unsupervised learning, a linear classifier is further trained on top of the frozen backbone and a global average pooling layer. In our experiments, the linear classifier is trained for 100 epochs using a SGD optimizer in a cosine decay schedule, where the base learning rate is 30, weight decay is 0, momentum is 0.9 and batch size is 128. After training, we perform the evaluation on the center cropped images in the evaluation/test set. In addition, to avoid the randomness of single run and make the comparison more fair, we run three times for each method (*i.e.,* unsupervised pre-training for three times) and report the top-1 mean accuracy as well as 95% confidence interval (*i.e.,* linear evaluation) for all comparison methods.

Table 1: Linear evaluation (top-1 mean accuracy) on four small benchmark datasets, all at 95% confidence intervals. Evaluation is on a single center crop. All unsupervised methods are trained 1000 epochs on CIFAR-10/100 and STL-10, and 200 epochs on ImageNet-100, all of which are trained for three times. In addition, SimCLR, SimSiam and Trip-ROMA are further trained 1000 epochs on ImageNet-100 for once. *Neg. pair* indicates whether negative pairs are used. *Stop. and Pred.* denotes whether an additional predictor network and stop-gradient are used.

| DataSet | Image Size | Method | Epochs | Batch Size | Backbone | Neg. Pair | Stop. Pred. | Acc. (%) |
|---|---|---|---|---|---|---|---|---|
| *CIFAR-10* | 32×32 | SimCLR (Chen et al., 2020a) | 1000 | 512 | ResNet-18 | ✓ | | $91.45_{\pm 0.18}$ |
| | | **SimCLR+ROMA** | 1000 | 512 | ResNet-18 | ✓ | | $91.93_{\pm 0.09}$ (↑ 0.48) |
| | | MoCo v2 (Chen et al., 2020b) | 1000 | 512 | ResNet-18 | ✓ | | $89.46_{\pm 0.32}$ |
| | | SimSiam (Chen & He, 2021) | 1000 | 512 | ResNet-18 | | ✓ | $91.12_{\pm 0.12}$ |
| | | **SimSiam+ROMA** | 1000 | 512 | ResNet-18 | | ✓ | $91.61_{\pm 0.09}$ (↑ 0.49) |
| | | **Trip** | 1000 | 64 | ResNet-18 | ✓ | | $\mathbf{92.06_{\pm 0.21}}$ |
| | | **Trip-ROMA** | 1000 | 64 | ResNet-18 | ✓ | | $\mathbf{92.29_{\pm 0.18}}$ |
| *CIFAR-100* | 32×32 | SimCLR (Chen et al., 2020a) | 1000 | 512 | ResNet-18 | ✓ | | $62.66_{\pm 0.12}$ |
| | | **SimCLR+ROMA** | 1000 | 512 | ResNet-18 | ✓ | | $63.05_{\pm 0.39}$ (↑ 0.39) |
| | | MoCo v2 (Chen et al., 2020b) | 1000 | 512 | ResNet-18 | ✓ | | $59.77_{\pm 0.69}$ |
| | | SimSiam (Chen & He, 2021) | 1000 | 512 | ResNet-18 | | ✓ | $64.04_{\pm 0.32}$ |
| | | **SimSiam+ROMA** | 1000 | 512 | ResNet-18 | | ✓ | $\mathbf{66.35_{\pm 0.36}}$ (↑ 2.31) |
| | | **Trip** | 1000 | 64 | ResNet-18 | ✓ | | $66.06_{\pm 0.36}$ |
| | | **Trip-ROMA** | 1000 | 64 | ResNet-18 | ✓ | | $\mathbf{66.32_{\pm 0.32}}$ |
| *STL-10* | 64×64 | SimCLR (Chen et al., 2020a) | 1000 | 512 | ResNet-50 | ✓ | | $87.14_{\pm 0.19}$ |
| | | **SimCLR+ROMA** | 1000 | 512 | ResNet-50 | ✓ | | $87.28_{\pm 0.28}$ (↑ 0.14) |
| | | MoCo v2 (Chen et al., 2020b) | 1000 | 512 | ResNet-50 | ✓ | | $86.69_{\pm 0.56}$ |
| | | SimSiam (Chen & He, 2021) | 1000 | 512 | ResNet-50 | | ✓ | $85.94_{\pm 0.82}$ |
| | | **SimSiam+ROMA** | 1000 | 512 | ResNet-50 | | ✓ | $86.18_{\pm 0.44}$ (↑ 0.24) |
| | | **Trip** | 1000 | 128 | ResNet-50 | ✓ | | $\mathbf{87.37_{\pm 0.42}}$ |
| | | **Trip-ROMA** | 1000 | 128 | ResNet-50 | ✓ | | $\mathbf{87.80_{\pm 0.27}}$ |
| *ImageNet-100* | 224×224 | SimCLR (Chen et al., 2020a) | 200 | 256 | ResNet-50 | ✓ | | $76.15_{\pm 0.36}$ |
| | | **SimCLR+ROMA** | 200 | 256 | ResNet-50 | ✓ | | $76.53_{\pm 0.22}$ (↑ 0.38) |
| | | MoCo v2 (Chen et al., 2020b) | 200 | 256 | ResNet-50 | ✓ | | $68.59_{\pm 0.51}$ |
| | | SimSiam (Chen & He, 2021) | 200 | 256 | ResNet-50 | | ✓ | $75.15_{\pm 0.58}$ |
| | | **SimSiam+ROMA** | 200 | 256 | ResNet-50 | | ✓ | $76.08_{\pm 0.44}$ (↑ 0.93) |
| | | **Trip** | 200 | 128 | ResNet-50 | ✓ | | $\mathbf{78.62_{\pm 0.11}}$ |
| | | **Trip-ROMA** | 200 | 128 | ResNet-50 | ✓ | | $\mathbf{80.21_{\pm 0.27}}$ |
| | | SimCLR (Chen et al., 2020a) | 1000 | 256 | ResNet-50 | ✓ | | 79.98 |
| | | **SimCLR+ROMA** | 1000 | 256 | ResNet-50 | ✓ | | 80.98 (↑ 1.00) |
| | | SimSiam (Chen & He, 2021) | 1000 | 256 | ResNet-50 | | ✓ | 81.72 |
| | | **SimSiam+ROMA** | 1000 | 256 | ResNet-50 | | ✓ | 82.10 (↑ 0.38) |
| | | **Trip** | 1000 | 128 | ResNet-50 | ✓ | | **81.97** |
| | | **Trip-ROMA** | 1000 | 128 | ResNet-50 | ✓ | | **83.05** |

## 5 Experiments

### 5.1 Comparison with the State of the Art

We thoroughly compare the proposed methods with the closely related state-of-the-art methods, *i.e.,* Sim-CLR (Chen et al., 2020a), MoCo v2 (Chen et al., 2020b) and SimSiam (Chen & He, 2021), on four benchmark datasets, including *CIFAR-10*, *CIFAR-100*, *STL-10* and *ImageNet-100*. Specifically, two variants of Trip-ROMA are constructed by introducing the random mapping strategy into SimCLR and SimSiam, named *SimCLR+ROMA* and *SimSiam+ROMA*, respectively. In addition, to verify the effectiveness of the proposed triplet-based loss in Eq. (1), another variant of Trip-ROMA is also constructed by removing the random map-pings, *i.e., Trip. For fairness, we reimplement all the comparison methods into the same framework being faithful to the original paper.* Following the literature (Tian et al., 2019; Chen et al., 2020a; Chen & He, 2021), we adopt a *ResNet-50* backbone on STL-10 and ImageNet-100, and adopt a *ResNet-18* backbone on CIFAR-10 and CIFAR-100 for all comparison methods. The linear evaluation results are reported in Table 1.

**Effectiveness of the triplet-based loss.** One of our concerns is how to effectively utilize negatives in SSL. From Table 1, we see that *Trip* (which only uses the proposed triplet-based loss) can achieve remarkable performance on all datasets. For example, on CIFAR-10, *Trip* can obtain 0.61%, 2.60% and 0.94% improvements over SimCLR, MoCo v2 and SimSiam, respectively. More importantly, *Trip* adopts a much smaller batch size (64), compared to other competitors (512). Similarly, on ImageNet-100, *Trip* can even gain 2.47%, 10.03% and 3.47% improvements over SimCLR, MoCo v2 and SimSiam, respectively. It verifies that negatives are still important and one is sufficient, *i.e.,* triplet is sufficient. Also, the above concern, *i.e.,* the first question in the introduction, has been answered.

**Effectiveness of random mappings.** As seen in Table 1, SimCLR+ROMA consistently performs better than SimCLR, and obtains 0.48%, 0.39%, 0.14% and 0.38% improvements over SimCLR on four datasets, respectively. Similarly, SimSiam+ROMA can also consistently performs better than SimSiam, and achieves 0.49%, 2.31%, 0.24% and 0.93% improvements, respectively. This successfully demonstrates the effectiveness and generality of the proposed random mapping strategy. Meanwhile, the second question in the introduction can also be answered.

**Effectiveness of Trip-ROMA.** As expected, Trip-ROMA, a combination of the triplet-based loss and random mapping strategy, achieves remarkable performance on all datasets, and performs much better than other competitors. For example, Trip-ROMA achieves a new state-of-the-art result of 92.29% on CIFAR-10 with a much smaller batch size of 64, which is 0.84%, 2.83% and 1.17% higher than SimCLR, MoCo v2 and SimSiam with a batch size of 512, respectively. Similarly, on CIFAR-100, Trip-ROMA still performs better than other competitors, gaining 3.66%, 6.55% and 2.28% improvements over SimCLR, MoCo v2 and SimSiam, respectively. Specifically, on the more challenging ImageNet-100, Trip-ROMA still gain significantly improvements over SimCLR, MoCo v2 and SimSiam by 4.06%, 11.62% and 5.06%, respectively, even with a smaller batch size of 128.

**Longer training process.** We also compare Trip-ROMA with SimCLR and SimSiam in a longer training process (1000 epochs) on ImageNet-100. As shown in Fig. 3(a), in the first 400 epochs, Trip-ROMA obtains similar accuracy as SimCLR. However, after 400 epochs, Trip-ROMA improves significantly and achieves 2.05%, 3.34%, 3.07% improvements over SimCLR at the 600th, 800th, 1000th epochs, respectively. Although SimSiam can converge faster than Trip-ROMA in the initial stage, its performance will becomes saturation or even slightly degraded after 400 epochs. On the contrary, the performance of Trip-ROMA can steadily improve with longer training. Their results trained for 1000 epochs are also reported in Table 1. As seen, Trip-ROMA obtains a new state-of-the-art result of 83.05% on ImageNet-100, which is 3.07% and 1.33% higher than SimCLR and SimSiam, respectively. This verifies that Trip-ROMA has a strong ability to continuously learn powerful representations from randomness with longer training.

**Performance on large-scale data.** Although our focus in this work is on small data, we also perform Trip and Trip-ROMA on a large-scale dataset, *i.e.,* ImageNet-1K (Deng et al., 2009) to fully investigate the performance of Trip and Trip-ROMA. Different from the small datasets, as a large-scale dataset, ImageNet-1K needs more computing resources to put effort into efficiently tuning the hyper-parameters and training tricks. Unfortunately, we are not able to have access to sufficient computing resources. Therefore, we simply

Table 2: Linear evaluation (top-1 accuracy) on ImageNet-1K. All the methods are trained with a *ResNet-50*. * means that the multi-crop strategy is used.

| Method | Epochs | Batch Size | Acc. (%) |
|---|---|---|---|
| NPID (Wu et al., 2018) | 200 | 256 | 54.0 |
| LA (Zhuang et al., 2019) | 200 | 128 | 60.2 |
| MoCo (He et al., 2020) | 200 | 256 | 60.6 |
| SeLa (Asano et al., 2019) | 200 | 256 | 61.5 |
| PIRL (Misra & Maaten, 2020) | 200 | 1024 | 63.6 |
| CPC v2 (Hénaff et al., 2020) | 200 | 512 | 63.8 |
| SimCLR (Chen et al., 2020a) | 200 | 256 | 64.3 |
| SimCLR (Chen et al., 2020a) | 200 | 4096 | 66.8 |
| MoCo v2 (Chen et al., 2020b) | 200 | 256 | 67.5 |
| SwAV (Caron et al., 2020) | 200 | 4096 | 69.1 |
| SwAV* (Caron et al., 2020) | 200 | 4096 | **72.7** |
| SimSiam (Chen & He, 2021) | 200 | 256 | 70.0 |
| BYOL (Grill et al., 2020) | 200 | 4096 | 70.6 |
| **Trip** | 200 | 256 | **69.9** |
| **Trip-ROMA** | 200 | 256 | **70.2** |
| **Trip-ROMA*** | 200 | 256 | **71.1** |

Table 3: $k$-NN evaluation (top-1 mean accuracy) on *mini*ImageNet, CIFAR-100FS and FC100 under the unsupervised few-shot setting. ‡ denotes that the results are quoted from the original papers.

| Type | Method | Backbone | *mini*ImageNet | | CIFAR-100FS | | FC100 | |
|---|---|---|---|---|---|---|---|---|
| | | | 5w1s | 5w5s | 5w1s | 5w5s | 5w1s | 5w5s |
| Supervised FSL | ProtoNet (Snell et al., 2017) | ResNet-12 | 57.10 | 74.20 | 57.64 | 81.24 | 35.16 | 49.34 |
| Unsupervised FSL | UBC-FSL (Chen et al., 2021)‡ | ResNet-12 | 47.8 | 68.5 | - | - | - | - |
| | No-labels (Bharti et al., 2020)‡ | ResNet-50 | 50.1 | 60.1 | 53.0 | 62.5 | **37.1** | 43.4 |
| | UBC-FSL (Chen et al., 2021)‡ | ResNet-50 | 56.2 | 75.4 | - | - | - | - |
| Self-supervised Learning | SimSiam (Chen & He, 2021) | ResNet-12 | 52.76 | 73.30 | 50.78 | 69.54 | 35.11 | 46.93 |
| | SimCLR (Chen et al., 2020a) | ResNet-12 | 57.13 | 74.54 | 55.27 | 73.56 | 35.01 | 47.26 |
| | **Trip-ROMA** | ResNet-12 | **58.02** | **76.41** | 55.87 | **74.00** | 36.30 | **49.06** |

follow the latest works (Chen et al., 2020a; Grill et al., 2020; Caron et al., 2021; Bardes et al., 2022), adopt a *ResNet-50* backbone on ImageNet-1K and introduce some training tricks, *e.g.,* momentum encoder (Grill et al., 2020) and multi-crop (Caron et al., 2020) into the training process. In addition, we also perform a variant of Trip-ROMA without using the multi-crop strategy. As seen in Table 2, Trip-ROMA* obtains a top-1 accuracy of 71.1% with a small batch size of 256 under a 200-epoch training setting, which is better than MoCo (He et al., 2020), SimCLR (Chen et al., 2020a), SimSiam (Chen & He, 2021) and BYOL (Grill et al., 2020), and is very close to SwAV* of using a much larger batch size of 4096. This successfully demonstrates the effectiveness and applicability of the proposed methods on the large-scale dataset.

## 5.2 Performance on Unsupervised FSL Tasks

We further apply Trip-ROMA to unsupervised few-shot learning tasks to investigate its effectiveness on the scenario of few-shot learning (FSL). Following the FSL literature (Lee et al., 2019; Chen et al., 2019), we adopt ResNet-12 as the backbone and set the input image size to $84 \times 84$. At the training stage, we perform unsupervised pre-training on each data (*i.e., mini*ImageNet, CIFAR-100FS and FC100) for 1000 epochs by using Trip-ROMA, SimCLR and SimSiam, respectively. At the test stage, we use a $k$-NN evaluation, *i.e.,* a cosine similarity based ProtoNet (Snell et al., 2017), over 3000 5-way 1-shot / 5-shot test tasks and report the mean top-1 accuracy.

As shown in Table 3, compared with the closely related SSL methods (*i.e.,* SimCLR and SimSiam), Trip-ROMA achieves the best results on all three datasets with a $k$-NN evaluation. Specifically, on *mini*ImageNet, Trip-ROMA gains 1.87% and 3.11% improvements over SimCLR and SimSiam under the 5-shot setting,

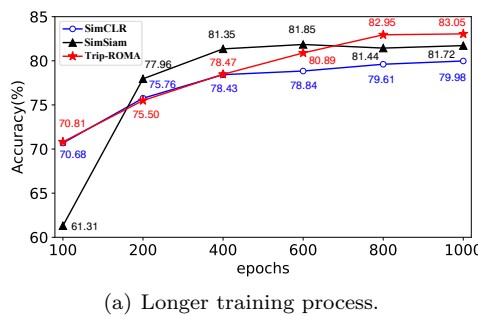 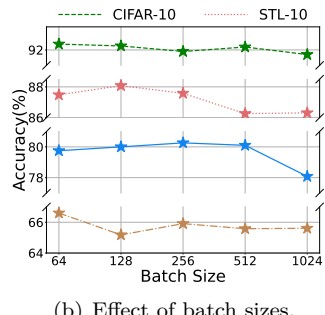 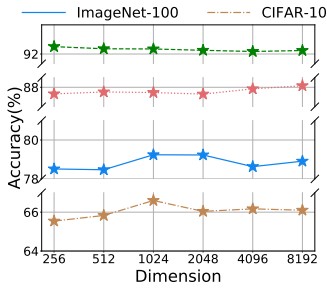

| (a) Longer training process. | (b) Effect of batch sizes. | (c) Random output dimensions. |

Figure 3: (a) Performance on ImageNet-100 for a longer training, *i.e.,* 1000 epochs. Here each point is an intermediate linear evaluation (top-1 accuracy) result during the 1000-epoch training. (b) Effect of different batch sizes for Trip-ROMA. (c) Effect of random output dimensions for Trip-ROMA.

Table 4: Effects of different loss functions.

|  | Triplet | CE | Trip |
|---|---|---|---|
| CIFAR-10 | 91.67 | 91.71 | **92.29** |
| ImageNet-100 | 78.12 | 77.28 | **80.21** |

Table 5: Effects of different random frequencies.

|  | NoRandom | 1Batch | 1Epoch | 10Epoch |
|---|---|---|---|---|
| CIFAR-10 | 91.85 | 92.19 | 92.26 | **92.29** |
| ImageNet-100 | 77.14 | 79.11 | **80.21** | 79.07 |

Table 6: Effects of using different random strategies.

|  | Bernoulli | Uniform | Normal |
|---|---|---|---|
| CIFAR-10 | 91.28 | 91.57 | **92.29** |
| ImageNet-100 | 77.66 | 78.67 | **80.21** |

Table 7: Effects of using the same batch size for all methods on ImageNet-100.

| Method | Epochs | BatchSize | Backbone | Acc.(%) |
|---|---|---|---|---|
| SimCLR | 200 | 128 | ResNet-50 | 76.85 |
| MoCo v2 | 200 | 128 | ResNet-50 | 69.32 |
| SimSiam | 200 | 128 | ResNet-50 | 74.12 |
| **Trip** | 200 | 128 | ResNet-50 | **78.62** |
| **Trip-ROMA** | 200 | 128 | ResNet-50 | **80.21** |

respectively. Also, compared with the existing unsupervised FSL methods, *i.e.,* UBC-FSL (Chen et al., 2021) and No-labels (Bharti et al., 2020), Trip-ROMA is still able to obtain significant improvements no matter under the 1-shot nor 5-shot settings on both *mini*ImageNet and CIFAR-100FS. No-labels performs better than Trip-ROMA only on FC100 under the 1-shot setting, but note that No-labels uses a much deeper backbone than Trip-ROMA. In addition, it is interesting see that Trip-ROMA, an SSL method, has already been able to achieve much better results over ProtoNet (Snell et al., 2017), a supervised FSL method, on *mini*ImageNet under the *k*-NN evaluation criterion. This reveals that the SSL methods alone are greatly promising in the field of FSL.

### 5.3 Ablation Studies

**Batch size.** For contrastive SSL methods, *e.g.,* SimCLR (Chen et al., 2020a), the excellent performance is largely dependent on a large batch size, while the performance will suffer drops when batch size is reduced. In contrast, our Trip-ROMA is more robust to small batch sizes, like some non-contrastive SSL methods, *e.g.,* SimSiam (Chen & He, 2021). To empirically verify this point, we train Trip-ROMA on four datasets using different batch sizes from 64 to 1024. As seen in Fig. 3(b), Trip-ROMA works very well with a small batch size on all datasets, especially on CIFAR-10 and CIFAR-100. When using a batch size of 128 (or 256), Trip-ROMA performs the best on STL-10 (or ImageNet-100).

In addition, to further verify the superiority on a small batch size of the proposed methods, we conduct an experiment on ImageNet-100 by using the same small batch size of 128. From Table 7, we can see that SimCLR, MoCo v2 can maintain the results, but SimSiam will lose 1.03% performance compared to their

results in Table 1. In this sense, performance improvements of the proposed Trip and Trip-ROMA over these existing methods are robust and significant.

**Random dimension.** The output dimension of the random projection matrix, *i.e.,* the dimension of the random subspace, is also an important factor in Trip-ROMA. Note that the feature dimension is usually set as 2048 in the literature. Clearly, we can not only randomly project samples into a lower-dimensional subspace (*e.g.,* 256), but also project them into a higher-dimensional subspace (*e.g.,* 8192). To systematically study the impacts of different random dimensions, we conduct experiments on four benchmark datasets by varying the random output dimensions. As seen in Fig. 3(c), on different datasets, the impacts of output dimensions are relatively different. For example, on STL-10, a higher-dimensional projection performs better, but the difference is somewhat limited. In contrast, on CIFAR-100 and ImageNet-100, a 1024-dimensional projection performs the best. Intuitively, such an appropriate dimensionality reduction, *i.e.,* 2048×1024, could reduce some redundant features, benefitting the final performance. In general, a random projection matrix with the size of 2048×1024 can obtain stable good results for Trip-ROMA.

**Loss function.** Because there are two parts in our objective loss function in Eq. (3), *i.e.,* a triplet loss and a binary cross-entropy (CE) loss, it will be interesting to analyze the influence of each individual part. Without loss of generality, we conduct experiments on both CIFAR-10 and ImageNet-100 with the same settings in Section 5.1. In particular, three variants of Trip-ROMA are constructed, *i.e., Triplet*, *CE* and *Trip*, where *Triplet* means only the triplet loss is used, *CE* indicates only binary cross-entropy loss is used, and *Trip* means both parts are used. Note that all the three variants still employ the random mapping strategy. From Table 4, we can see that (1) each individual loss can achieve competitive results, compared to the state-of-the-art methods (see Table 1); (2) Triplet+CE performs much better than each individual loss alone (*i.e.,* Triplet or CE). As have been explained in Section 3.1, the triplet-based cross-entropy loss can be regarded as a supplementary to the standard triplet loss, both of which benefit the final performance of Trip-ROMA.

**Random frequency.** The frequency of random mapping determines how much knowledge can our method acquire from the latent random subspaces. An inappropriate random frequency may cause our method to fail to acquire enough knowledge or waste too much time in a random subspace. Therefore, we compare the performance of using different frequencies, *i.e.,* using no random (*NoRandom*), using random every mini batch (*1Batch*), using random every epoch (*1Epoch*) and using random every 10 epochs (*10Epoch*), on both CIFAR-10 and ImageNet-100. As seen in Table 5, compared to *NoRandom*, all three kinds of random frequencies can boost the performance, which further verifies the effectiveness of the random mapping strategy. In addition, we can see that Trip-ROMA is not significantly sensitive to the random frequency. For example, on CIFAR-10, using random every 10 epochs (*10Epoch*), *i.e.,* a lower frequency, performs the best, while using random every epoch (*1Epoch*), a higher frequency, achieves the highest accuracy on ImageNet-100. Therefore, in our other experiments, we use *1Epoch* frequency for ImageNet-100 and *mini*ImageNet, and *10Epoch* frequency for other five datasets which are trained for 1000 epochs.

**Random strategy.** We further explore the impacts of different random strategies. Specifically, three distributions are employed, including Bernoulli distribution (*Bernoulli*), uniform distribution of $U(-1, 1)$ (*Uniform*), and standard normal distribution (*Normal*), where the first is a discrete distribution, while the later two are continuous distributions. For example, if we employ the standard normal distribution to generate a random projection matrix $L$, it means that all entries of $L$ are identically and independently sampled from a standard normal distribution. Table 6 presents the results on CIFAR-10 and ImageNet-100. From the results, we can observe that the normal distribution strategy outperforms *Bernoulli* and *Uniform* strategies on both of datasets. In addition, we note that the continuous random strategy (*i.e.,* the Uniform and Normal distributions) performs better than the discrete random strategy (*i.e.,* the Bernoulli distribution) to some extent, which implies that the random continuous subspace is more conductive to learn more robust representations.

## 6   Conclusion

In this paper, we present Trip-ROMA, a simple and effective SSL method for unsupervised representation learning and unsupervised few-shot learning. By using a simple triplet-based loss and a random mapping

strategy, Trip-ROMA can achieve new state-of-the-art results on seven small datasets and a large-scale dataset. From the results, we have the following findings: (1) Contrary to the recent trend of completely discarding the negative pairs in SSL, negative examples are still important but one is sufficient with a properly designed loss function; (2) Unsupervised representation learning can benefit significantly from randomness, such as using random mappings. We hope our work will draw the community to rethinking the impact of negative examples and exploiting randomness for benefits. In the future, we will explore how to effectively accelerate the convergence of Trip-ROMA.

**Acknowledgements.**    This work is supported in part by the Science and Technology Innovation 2030 New Generation Artificial Intelligence Major Project (2021ZD0113303), the National Natural Science Foundation of China (62106100, 62192783, 62276128), Jiangsu Natural Science Foundation (BK20221441), the Collaborative Innovation Center of Novel Software Technology and Industrialization, and Jiangsu Provincial Double-Innovation Doctor Program (JSSCBS20210021).

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

## A    Details of Datasets

As mentioned in the main paper, the major experiments of this paper are conducted on seven small benchmark datasets, *i.e.,* CIFAR-10 (Krizhevsky et al., 2009), CIFAR-100 (Krizhevsky et al., 2009), STL-10 (Coates et al., 2011), ImageNet-100 (Tian et al., 2019), *mini*ImageNet (Vinyals et al., 2016), CIFAR-100FS (Bharti et al., 2020), and FC100 (Oreshkin et al., 2018). A large-scale benchmark dataset, *i.e.*, ImageNet-1K (Deng et al., 2009), is also used. The details of these datasets are as follows:

- **CIFAR-10** consists of 60000 $32 \times 32$ colour images in 10 classes, in which there are 6000 images in each class. Following the original splits, we take 50000 and 10000 images for training (without labels) and test, respectively.

- **CIFAR-100** has a total number of 100 classes, containing 500 training images and 100 test images per class. We take 50000 images ignoring the labels for training, and the remaining images for test.

- **STL-10** contains 10 classes, where there are $100,000$ $96 \times 96$ unlabeled training images, and 500 labeled training images and 800 test images in each class.

- **ImageNet-100** is a subset of ImageNet-1K with 100 well-balanced classes and a total number of 0.13 million images, which is first introduced in (Tian et al., 2019).

- **mini*ImageNet** is a popular benchmark dataset in the filed of few-shot learning (FSL), which contains 100 classes selected from ImageNet-1K, each class has 600 images with an image size of $84 \times 84$.

- **CIFAR-100FS** is a version for few-shot learning conducted based on the CIFAR-100 dataset (Krizhevsky et al., 2009). It also contains 100 classes like CIFAR-100, where 64, 16 and 20 classes are used for training, validation and test, respectively. In addition, because the average inter-class similarity of this dataset is generally high and the image resolution of $32 \times 32$ is low, CIFAR-100FS is generally a challenging data in FSL.

- **FC100** is another version built on CIFAR-100 for FSL with a new splits that is different from CIFAR-100FS. Specifically, 20 high-level super classes are divided into 12, 4 and 4 classes, which are corresponding to 60, 20 and 20 low-level specific classes, for training, validation and test, respectively.

- **ImageNet-1K** is a widely-used large-scale dataset in the field of image classification. It contains 1000 classes, with 1.2 million images for training and 50000 images for validation. All images are resized to $224 \times 224$ for unsupervised training without labels.

## B    Pseudo-code of Trip-ROMA

The pseudo-code of the proposed Trip-ROMA is shown in Algorithm 1.

---

**Algorithm 1** Trip-ROMA Pseudocode, PyTorch-like

---

```
1  # f: backbone + projection mlp
2  for X, X_ in loader:                                  # load two samples
3      x1, x2, x3 = aut(X), aug(X), aug (X_)             # random augmentation, obtaining triplet
4      z1, z2, z3 = f(x1), f(x2), f(x3)                  # projection, N-by-D
5      random_matrix = randn(D, D)                       # random matrix: D-by-D
6      loss = L(z1, z2, z3, random_matrix)               # loss
7      loss.backward()                                   # backward
8      update(f)                                         # SGD update
9
10 def L(z1, z2, z3, random_matrix):
11     z1 = normalize(z1 @ random_matrix, dim=1)         # random projection and normalize
12     z2 = normalize(z2 @ random_matrix, dim=1)
13     z3 = normalize(z3 @ random_matrix, dim=1)
14
15     pos_sim = (z1 * z2).sum(dim=1)                     # positive pair similarity
16     neg_sim = (z1 * z3).sum(dim=1)                     # negative pair similarity
17
18     Triplet_loss = clamp(matgin + neg_sim - pos_sim, min=0.)    # Triplet loss
19
20     logits = cat([pos_sim, neg_sim]) / temperature    # scale by temperature
21     labels = [1, 0]                                    # label
22     CE_loss = cross_entropy(logits, labels)           # Cross entropy loss
23
24     return Triplet_loss + weight * CE_loss
```

---

## C Explanation for ROMA from Perturbation of the points

In addition to the explanation from the perturbation of the coordinates in Section 3.2 in the main paper, we can also explain the random projection, *i.e.,* ROMA, from another perspective, *i.e.,* perturbation of the points. Specifically, given a pair of points $\boldsymbol{u}$ and $\boldsymbol{v}$, we are able to add a small perturbation $\Delta\boldsymbol{u}, \Delta\boldsymbol{v} \in \mathbb{R}^d$ to $\boldsymbol{u}$ and $\boldsymbol{v}$, respectively. Conceptually, we can always generate them by $\Delta\boldsymbol{u} = M_1\boldsymbol{u}$ and $\Delta\boldsymbol{v} = M_2\boldsymbol{v}$, where $M_1$ and $M_2$ are two random matrices with a sufficiently small Frobenius norm. This is because the norm of $M_1\boldsymbol{u}$ can be proved to be upped bounded by the Frobenius norm of $M_1$ as follows: $\|M_1\boldsymbol{u}\|_2^2 = \boldsymbol{u}^\top M_1^\top M_1 \boldsymbol{u} \leq \lambda_{max}(M_1^\top M_1)\|\boldsymbol{u}\|_2^2 \leq trace(M_1^\top M_1)\|\boldsymbol{u}\|_2^2 = \|M_1\|_F^2\|\boldsymbol{u}\|_2^2$. This proof uses the property of Rayleigh quotient and the definition of Frobenius norm. Therefore, by using $M_1$ with a sufficiently small norm, we can generate a sufficiently small perturbation by $\Delta\boldsymbol{u} = M_1\boldsymbol{u}$. This proof applies to the case of $\Delta\boldsymbol{v} = M_2\boldsymbol{v}$. In this case, their cosine similarity can be expressed as

$$
\begin{aligned}
Sim(\boldsymbol{u} + \Delta\boldsymbol{u}, \boldsymbol{v} + \Delta\boldsymbol{v}) &= (\boldsymbol{u} + \Delta\boldsymbol{u})^\top(\boldsymbol{v} + \Delta\boldsymbol{v}) \\
&= \boldsymbol{u}^\top\boldsymbol{v} + \boldsymbol{u}^\top\Delta\boldsymbol{v} + \Delta\boldsymbol{u}^\top\boldsymbol{v} + \Delta\boldsymbol{u}^\top\Delta\boldsymbol{v} \\
&= \boldsymbol{u}^\top\boldsymbol{v} + \boldsymbol{u}^\top M_2\boldsymbol{v} + \boldsymbol{u}^\top M_1\boldsymbol{v} + \boldsymbol{u}^\top M_1^\top M_2\boldsymbol{v} \\
&= \boldsymbol{u}^\top(I + M_2 + M_1 + M_1^\top M_2)\boldsymbol{v} \\
&\quad \left(Let \ M \triangleq I + M_1 + M_2 + M_1^\top M_2\right) \\
&= \boldsymbol{u}^\top M\boldsymbol{v}.
\end{aligned}
\tag{7}
$$

As seen, when we interpret our random matrix $M$ as $(I + M_1 + M_2 + M_1^\top M_2)$, its effect can be thought of as adding a random perturbation to the points before evaluating their similarity as usual. This provides an explanation on why our insertion of random matrix $M$ could lead to better results.

## D Eigenvalue Spectrum Analysis

To gain more insight on the effect of the proposed random projection, *i.e.,* ROMA, we examine the eigenvalue spectrum of the covariance matrix computed with the feature representations before and after a random projection is applied. To be specific, we train Trip-ROMA on ImageNet-100 for 200 epochs with RestNet-50

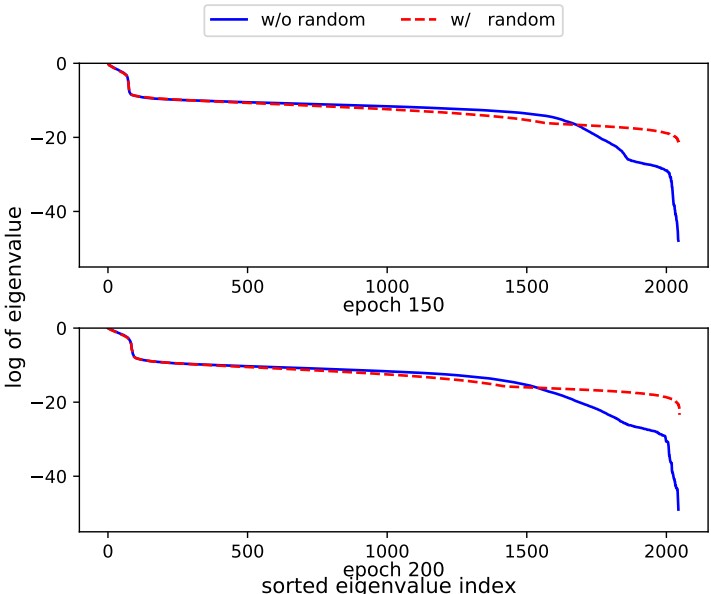

Figure 4: Evolution of eigenvalues of the covariance matrix computed with the feature representations learned by Trip-ROMA on ImageNet-100. The solid and dotted lines indicate the eigenvalues obtained before and after using a random projection, respectively.

and select the effectively trained models at the 150th and 200th epochs, respectively. With each model, we extract a 2048-dimensional feature vector for each training image, and then use all feature vectors to calculate the covariance matrix before and after a random projection is applied to them, respectively. For each of the two covariance matrices, its eigenvalues are sorted and linearly scaled to make the largest eigenvalue to be one for the ease of comparison. The resulted eigenvalue spectrums from the two covariance matrices are compared in Figure 4.

It is observed that after a random projection is applied, the eigenvalue spectrum becomes more balanced, that is, the rapid attenuation at the late stage of the spectrum has been well deferred. This change indicates that the variance of the feature representations along the directions of the corresponding eigenvectors is increased. This increased variance could help to reduce the odds that the relationship (*i.e.*, similar or not similar) of samples is satisfied by chance when it is measured in this new projected space.

## E   Implementation of Trip-ROMA on ImageNet-1K

Different from the small datasets, as a large-scale dataset, ImageNet-1K needs more computing resources to put effort into efficiently tuning the hyper-parameters and training tricks. Unfortunately, we are not able to have access to sufficient computing resources. Therefore, we follow the latest works (Chen et al., 2020a; Grill et al., 2020; Caron et al., 2021; Bardes et al., 2022), adopt a *ResNet-50* backbone on ImageNet-1K and introduce some training tricks, *e.g.,* momentum encoder (Grill et al., 2020) and multi-crop (Caron et al., 2020) into the training process. Specifically, there are two encoders used to extract features, one is directly updated by the gradients, and the other is momentum-updated by the former via the exponential moving average operation. As seen in Table 2, our Trip-ROMA achieves a top-1 accuracy of 71.1% with 200 training epochs, which is much better than the current state-of-the-art SSL methods, such as SimCLR (Chen et al., 2020a), SwAV (Caron et al., 2020), SimSiam (Chen & He, 2021) and BYOL (Grill et al., 2020).

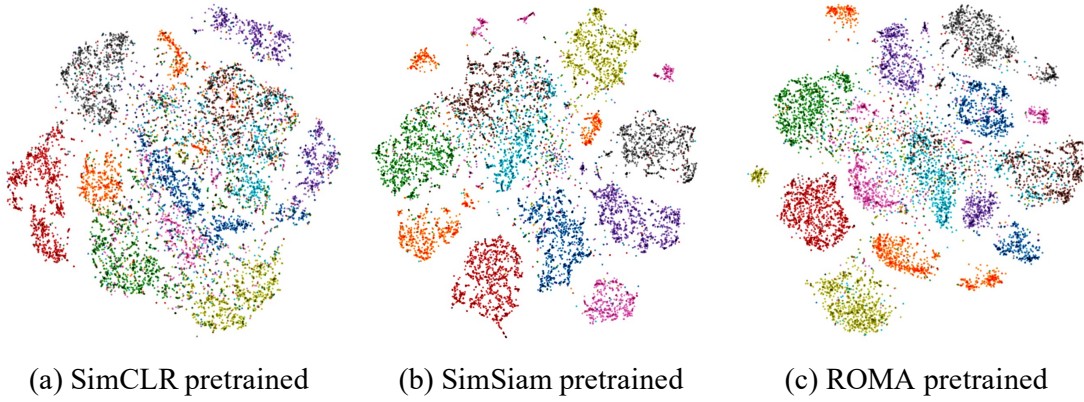

(a) SimCLR pretrained          (b) SimSiam pretrained          (c) ROMA pretrained

Figure 5: Feature visualization on CIFAR-10 with pretrained *ResNet-18* by SimCLR, SimSiam and Trip-ROMA, respectively. Each color indicates one class.

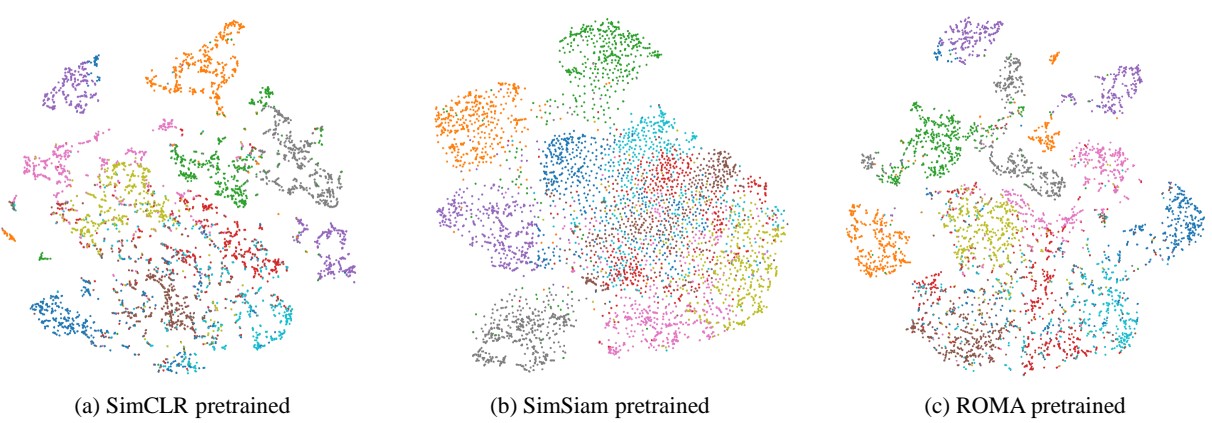

(a) SimCLR pretrained          (b) SimSiam pretrained          (c) ROMA pretrained

Figure 6: Feature visualization on STL-10 with pretrained *ResNet-50* by SimCLR, SimSiam and Trip-ROMA, respectively. Each color indicates one class.

## F  Feature Visualization

To further demonstrate the effectiveness of the proposed Trip-ROMA in a more intuitive way and show that the learned features are conducive to the classification, we visualize the feature spaces learnt by different methods on two datasets, *i.e.,* CIFAR-10 and STL-10, respectively.

As seen in Figure 5. First, three models are trained on CIFAR-10 with ResNet-18 by using SimCLR, SimSiam and Trip-ROMA, respectively. After that, all test samples in CIFAR-10 are represented accordingly and then are reduced to a two-dimensional space by t-SNE. As seen, the samples in the feature space learned by Trip-ROMA are more separable than both SimCLR and SimSiam, showing that Trip-ROMA can learn better feature representation.

Similarly, as seen in Figure 6, Trip-ROMA can also make the learnt features of different categories more separable than SimCLR and SimSiam, showing the effectiveness of Trip-ROMA in unsupervised representation learning. In addition, features learnt by SimSiam are more chaos than the features learnt by SimCLR and Trip-ROMA, showing that negative pairs are beneficial to learn more discriminative feature representations.

# G   Gradients Analysis

To further explain the effectiveness of the proposed loss function in Eq. (3), especially the complementary property of the two loss terms, we present the gradients analysis of this loss function. Recall that the formula of the proposed simple *triplet + binary cross-entropy loss* for a triplet is

$$\mathcal{L}_{ij} = \left[ \boldsymbol{z}_i^\top \boldsymbol{z}_j - \boldsymbol{z}_i^\top \tilde{\boldsymbol{z}}_i + 1 \right]_+ - \lambda \cdot \log \frac{\exp(\boldsymbol{z}_i^\top \tilde{\boldsymbol{z}}_i / \tau)}{\exp(\boldsymbol{z}_i^\top \tilde{\boldsymbol{z}}_i / \tau) + \exp(\boldsymbol{z}_i^\top \boldsymbol{z}_j / \tau)} \ . \tag{8}$$

To explain the effectiveness of each part in Eq. (8), we can calculate the gradients of $\mathcal{L}_{ij}$ with respect to the positive sample $\boldsymbol{z}_i$ as below:

$$\frac{\partial \mathcal{L}_{ij}}{\partial \boldsymbol{z}_i} = \begin{cases} (\boldsymbol{z}_j - \tilde{\boldsymbol{z}}_i) + \left( \frac{A}{A+B} \boldsymbol{z}_j - (\frac{\lambda}{\tau} - \frac{B}{A+B}) \tilde{\boldsymbol{z}}_i \right) & \text{if } \boldsymbol{z}_i^\top \tilde{\boldsymbol{z}}_i < \boldsymbol{z}_i^\top \boldsymbol{z}_j + 1 \\ \frac{A}{A+B} \boldsymbol{z}_j - (\frac{\lambda}{\tau} - \frac{B}{A+B}) \tilde{\boldsymbol{z}}_i & \text{otherwise} , \end{cases} \tag{9}$$

where $A = \exp(\boldsymbol{z}_i^\top \boldsymbol{z}_j / \tau)$ and $B = \exp(\boldsymbol{z}_i^\top \tilde{\boldsymbol{z}}_i / \tau)$. From Eq. (9), we have the following observations: (1) the gradient of the cross-entropy loss with respect to $\boldsymbol{z}_i$ has the same formula as the triplet loss, which means that these two loss items are complementary; (2) when the gradient of the triplet loss is zero, we still have the gradient of the binary cross-entropy loss to optimize the model.

