# OpenReview forum: "Trip-ROMA: Self-Supervised Learning with Triplets and Random Mappings"
_TMLR — Accepted by TMLR_

### Review · Reviewer_HcGJ · 2023-01-06

**Summary Of Contributions:**

This paper introduces observations related to contrastive self-supervised learning and a new approach that can reduce the training cost of the contrastive SSL, yet improve the quality of learned representations. Specifically, they show two observations: (i) the simple triplet loss plus cross-entropy loss, which requires only one negative sample per one positive sample, can achieve comparable or better performance than the existing contrastive SSL approaches requiring many negative samples, (ii) in addition to (i), projecting samples by a random matrix can also boost the performance. Their proposed approach combining (i) and (ii) can outperform many existing SSL methods. They conduct evaluations on diverse datasets and provide wide analysis including ablation studies.

**Audience:**

Yes

**Broader Impact Concerns:**

No.

**Claims And Evidence:**

Yes

**Requested Changes:**

I do not see any major concerns with this submission. The submission is already very well-written and has content that can be very impressive to the audience. My only concern is about the justification for why the proposed method is working so well. The existing papers' assumption that we require many negative samples to boost performance on contrastive SSL may not be true from this submission's result. But, there is not much empirical analysis on this point.

**Strengths And Weaknesses:**

### Strength
* Their observation is very interesting and practical. Although many existing SOTA methods are training SSL models with a large computational cost, the observation implies that we can dramatically reduce the cost by using their approaches.
* Their proposed random matrix projection, ROMA, is simple, yet useful for various SSL methods. This is also a great strength of this submission.
* The evaluation is extensive. The experiments cover almost everything I wanted to see.
* The submission is well-organized and easy to follow.

 ### Weaknesses
* The reason why simple triplet + cross-entropy loss is so effective on this task is not very well analyzed. I think it is a very interesting observation. But, if there is any analysis to understand the reason, the paper will be more solid.

---

> ### Author Response · Authors · 2023-01-17
> **Authors Response to Reviewer HcGJ**
>
> We appreciate the reviewer's insightful comments and positive support. We carefully address the main concern as below.
>
> **Q: The reason why simple triplet + cross-entropy loss is so effective on this task is not very well analyzed. I think it is a very interesting observation. But, if there is any analysis to understand the reason, the paper will be more solid.**
>
>
> Thanks for this constructive comment. We explain this point from three folds as below:
>
> (1). **Gradients Analysis**. As seen from Section G in the appendix, we have analyzed the gradients of the proposed triplet + cross-entropy loss. For convenience, we quote this part to further explain the effectiveness the proposed loss from the perspective of gradients.
>
>
> We first recall the formula of the proposed *triplet + binary cross-entropy loss* for a triplet as below:
>
> $$\mathcal{L}_{ij}=\lbrack z^\top_i z_j-z^\top_i\hat{z_i}+1\rbrack_\dotplus- \lambda\cdot\log\frac{\exp(z^\top_i\hat{z_i}/\tau)}{\exp(z^\top_i\hat{z_i}/\tau)+\exp(z^\top_i z_j/\tau)},$$
>
> where $z_i$ and $\hat{z_i}$ are the positive examples and $z_j$ is a negative example. Next, we can calculate the gradients of $\mathcal{L}_{ij}$ with respect to the positive sample $z_i$ as below:
>
> $$\frac{\partial\mathcal{L}_{ij}}{\partial z_i} = (z_j-\hat{z_i}) + \big(\frac{A}{A+B}z_j -(\frac{\lambda}{\tau}-\frac{B}{A+B})\hat{z_i}\big ) \quad \text{if } z^\top_i\hat{z_i}<z^\top_i z_j+1
> $$
>
> $$\frac{\partial\mathcal{L}_{ij}}{\partial z_i} = \frac{A}{A+B}z_j -(\frac{\lambda}{\tau}-\frac{B}{A+B})\hat{z_i}  \quad \text{otherwise},$$
>
> where $A=\exp(z^\top_i z_j/\tau)$ and $B=\exp(z^\top_i\hat{z_i}/\tau)$. From this equation, we have the following observations: (1) the gradient of the cross-entropy loss with respect to $z_i$ has the same formula as the triplet loss, which means that these two loss items are complementary; (2) when the gradient of the triplet loss is zero, we still have the gradient of the binary cross-entropy loss to optimize the model. In summary, the triplet loss and binary cross-entropy loss are complementary and promoted with each other.
>
> In addition, the above analysis can also be experimentally supported by the results from Table 4 in the main paper. As seen in Table 4, no matter using only the cross-entropy (CE) loss or using only the Triplet loss shows competitive results on both CIFAR-10 and ImageNet-100. As expected, combining these two losses together, i.e., the proposed Trip loss, can obtain much better results.
>
>
> (2). **Improvements from randomness**. We need to highlight that the effectiveness of the proposed method also gains a lot from the proposed random mapping (ROMA) strategy to some extent. As seen in Table 1 in the main paper, especially on the ImageNet-100 dataset, SimCLR+ROAM and SimSiam+ROMA only gain $0.38 \\%$ and $0.93\\%$ performance improvement over SimCLR and SimSiam, respectively. In contrast, our Trip-ROMA can gains $1.59\\%$ performance improvements over the proposed Trip (i.e., the proposed triplet + binary cross-entropy loss). This shows that the proposed triplet + binary cross-entropy loss can gain superior performance with the help of the proposed ROMA strategy.
>
>
> (3). **Support from non-contrastive learning based methods**. The effectiveness and reasonability of only using one example for each positive pair in our proposed method can also be somewhat supported by the existing non-contrastive learning based methods, such as SimSiam and BYOL, which only use positive examples without using any negative examples. The effectiveness of these methods has been successfully demonstrated by the recent SSL works. The key difference between our method and these non-contrastive based method is that we use one more negative example for each pair of positive examples. The advantage of using one negative example is that we can naturally avoid the collapse problem suffered and tried to address by the non-contrastive based SSL methods. Importantly, the effectiveness of the proposed method is successfully demonstrated in our paper experimentally, especially in the small data regimes.

---

### Review · Reviewer_6v58 · 2023-02-10

**Summary Of Contributions:**

In this paper, the authors propose a method that employs a Triplet-based loss and a random mapping strategy (ROMA). For the Triplet-based loss. There are two parts: a triplet loss and a cross-entropy loss for binary classification. ROMA employs a random matrix, which can be seen as many linear projections for vectors. For each projection, ROMA brings two positive vectors closer and pushes away the negative vector. The experimental results show the proposed method can largely improve the performance of small datasets and FSL tasks.

**Audience:**

Yes

**Claims And Evidence:**

No

**Requested Changes:**

1. In page 1, "the existing SSL methods are mainly ... while paying less attention to small data." There are some existing SSL methods that work well on small datasets.

[1] ReSSL: Relational Self-Supervised Learning with Weak Augmentation NerualIPS 2020
[2] RSA: Reducing Semantic Shift from Aggressive Augmentations for Self-supervised Learning. NerualIPS 2021
[3] Whitening for Self-Supervised Representation Learning. ICML 20212.

2. On page 4, "τ is a temperature parameter that is set as 0.5 in our work." Hyper-parameters should be given in the experimental part.

3. In the ImageNet-1K experiments, the authors adopt a multi-crop strategy, which can significantly improve the performance and increase the training cost. I think the authors should mention it in the paper, not supplimentary. For a fair comparsion and showing improvements, I suggest the results on ImageNet-1K without multi-crop should be given. BTW, SwAV with multi-crop is 72.7 in the original paper.

4. I found the matrix in ROAM to be relatively large, and it is used in each pair calculation. Thus, the training time comparison is important for ROAM. In addition, the most improvements of ROAM are less than 0.5 in table 1, and some important experiments are missing, e.g., ImageNet-100 for 1000 epochs, and ImageNet-1K w/wo ROAM.

**Strengths And Weaknesses:**

Strengths:
1. The exploration of the triplet loss is important for reducing the sizes of batch-size and memory banks in SSL.
2. The overall writing is good and can be easily followed.
3. The article gives a lot of experimental results, especially ablation experiments.


Weaknesses:
1. Why is the combination of two triplet losses better than any one of them? I think this part should give more explanation.
2. Critical experiments are missing. For example, because the result in ImageNet-1K uses multi-crop (which is lower than SwAV in the original paper), the effectiveness of the proposed method on large datasets cannot be verified.
For more details, please check "Requested Changes."

---

### Review · Reviewer_zTHE · 2023-02-20

**Summary Of Contributions:**

The authors propose two main extensions to existing contrastive self-supervised learning methods. First, they experiment with the use of a triplet loss (Trip.) with a single negative, by opposition with methods using a large number of negative examples or methods using only positive examples. Second, they propose the introduction of random mappings (ROMA) added on top of the computation of the embeddings in SSL methods.
The authors study the impact of their method in linear evaluation and few-shot-learning settings on different ResNet architectures and various datasets. Through ablation studies, the authors claim that both Trip and ROMA bring significant improvements in SSL.

**Audience:**

Yes

**Broader Impact Concerns:**

The paper raises no appearent ethical concerns.

**Claims And Evidence:**

Yes

**Requested Changes:**

# Open questions
In the spirit of TMLR, I believe the provided methods and results can already interest some readers and the claims experiments are technically correct. However the work would be greatly strenghtened by addressing the following open questions.

1. Intuition for ROMA: The justification for ROMA is unclear to me and I would like to have more certainty that the observed effects are consistent with the explanation. First, it is unclear what the authors call "overfitting" or "by chance" in the triplet relation, and this argument could be made more expicit. Second, it would be good to know why the random projections approach is actually beneficial, as my intuition would be that adding triplet relationships on the higher-dimensional space should already imply some triplet relationships on the projected images. Clarifying this intuition would be important before publication. Additionally, the authors could complement their experiments with varying the dimension of the projected subspaces, or by using axis-aligned subspaces (i.e. subsampling the embedding coordinates).

2. Impact of $\lambda$ (eq. 1): $\lambda$ seems arbitrarily set to 8 in the experiments. The work would be strenghtened by an ablation on the impact of $\lambda$ including the edge cases where we have only the triplet loss ($\lambda=0$) or only the cross-entropy loss (would require the addition of e.g. a ($1-\lambda$ factor for the first term and set $\lambda=1$) would be valuable in assessing if both loss terms contribute.

3. Influence of batch size: Trip. is compared with other methods with larger batch size. It would be beneficial to compare when using the same batch size and see if the gains of Trip are more important. Alternatively, it would be interesting to know if the performance of competing methods decreasses when using a smaller batch size.

# Misc.
4. Introduction: I do not agree with the author's claim that the asymetric momentum encoder or stop-gradient approaches of BYOL or SimSiam correspond to a "sophisticated" design. Indeed, these approaches are conceptually simple and easy to implement. I would however agree that requiring less batch size and could be a positive for using only a batch-based triplet loss.

5. Please distinguish between `\citep` to add parentheses to the citations, or `\citet` when the citations can be integrated in the text. At the moment the uniform use of `\citet` hinders the reading.

6. Sec. 5.2: please remind the reader of what "FSL" means (few-shot learning).


**Strengths And Weaknesses:**

# Strengths
* The methods are clearly exposed.
* The experiments are extensive. I appreciate in particular the 3-fold repetition and the addition of standard errors in Table 1.

# Weaknesses
* The paper feels like a combination of two unrelated methods - the use of a batch-based triplet loss combined with cross-entropy on one side, and the use of random projections of the other. Combining these two unrelated methods make the assessment of the significance of these two contributions more difficult in experiments where the ablation of using only one of the two is not done (e.g.: "Longer training process).
* The intuition to ROMA is somewhat unclear (see Requested changes 1. below)
* Some additional experiments would strenghten the experimental section significantly by reducing the unstudied hyperparameters (e.g. influence of $\lambda$, and dimension of projected subspaces in ROMA; see points in "Requested changes").

---

### Decision · Action_Editors · 2023-05-04

**Recommendation:** Accept as is

**Comment:**

The results in this paper may encourage others to investigate approaches that require few negative examples, making for scalable learning procedures. Two of the reviewers had no hesitations in recommending acceptance after discussion with the authors. The third reviewer pointed out that in a particular setting, without multi-cropping, performance was below SOTA, and that the method was somewhat complex. Overall i dont think this is of sufficient concern to overrule the other two assessments. Even if the model is a little complex, it still may have useful insights that later papers can use in more elegant ways.

**Audience:**

Self-supervised learning is still a hot topic, particularly in computer vision.

**Claims And Evidence:**

Paper provides a large number of results to support their main points. In the discussion with reviewers additional results were provided which completely satisfied two of the three reviewers.